# Comprehensive generation, visualization, and reporting of quality control metrics for single-cell RNA sequencing data

Rui Hong[1,2,6], Yusuke Koga[1,2,6], Shruthi Bandyadka [1,3], Anastasia Leshchyk [1,2], Yichen Wang[2], Vidya Akavoor[2,4], Xinyun Cao[4], Irzam Sarfraz[2], Zhe Wang [1,2], Salam Alabdullatif[2], Frederick Jansen[4], Masanao Yajima [5], W. Evan Johnson [1,2] & Joshua D. Campbell [1,2✉]

Single-cell RNA sequencing (scRNA-seq) can be used to gain insights into cellular heterogeneity within complex tissues. However, various technical artifacts can be present in scRNA-seq data and should be assessed before performing downstream analyses. While several tools have been developed to perform individual quality control (QC) tasks, they are scattered in different packages across several programming environments. Here, to streamline the process of generating and visualizing QC metrics for scRNA-seq data, we built the SCTK-QC pipeline within the *singleCellTK* R package. The SCTK-QC workflow can import data from several single-cell platforms and preprocessing tools and includes steps for empty droplet detection, generation of standard QC metrics, prediction of doublets, and estimation of ambient RNA. It can run on the command line, within the R console, on the cloud platform or with an interactive graphical user interface. Overall, the SCTK-QC pipeline streamlines and standardizes the process of performing QC for scRNA-seq data.

[1] Bioinformatics Program, Boston University, Boston, MA, USA. [2] Section of Computational Biomedicine, Boston University School of Medicine, Boston, MA, USA. [3] Department of Biology, Boston University, Boston, MA, USA. [4] Software & Application Innovation Lab, Rafik B. Hariri Institute for Computing and Computational Science and Engineering, Boston, MA, USA. [5] Department of Mathematics and Statistics, Boston University, Boston, MA, USA. [6] These authors contributed equally: Rui Hong, Yusuke Koga. ✉email: camp@bu.edu

Single-cell RNA-sequencing (scRNA-seq) has been instrumental in providing detailed insights into cellular heterogeneity related to tissue development and disease pathogenesis[1]. With the advent of microfluidic devices, the transcriptome for thousands of individual cells can be measured in a single run[2]. These devices work by partitioning cells into droplets along with beads containing oligonucleotide primers with unique barcodes. Within each droplet, reverse transcription is initially used to create barcoded cDNA and then additional amplification steps are used to create final sequencing libraries depending on the protocol[3]. Other approaches such as SMART-seq2 and CEL-seq2 can be used to profile cells that have been sorted into 96- or 384-well plates[4,5]. Many of these protocols use unique molecular indices (UMIs) to barcode each individual mRNA molecule and correct for biases in amplification[6].

Despite the advances in scRNA-seq protocols, poor-quality cells can still be present in high-quality runs. Technical artifacts related to the cell dissociation process, cell encapsulation, library preparation, or sequencing can affect various aspects of data quality. Low quality cells need to be excluded and technical artifacts need to be systematically assessed in each sample before downstream analyses can be performed. We briefly describe five types of QC analyses and metrics that are commonly utilized in scRNA-seq data analysis: (1) Cells in which barcoding or amplification reactions were not successful will have lower numbers of UMIs and genes detected. Lower numbers of UMIs and detected genes can hinder downstream analyses such as clustering because the genes that are able to distinguish cell populations may not be adequately measured. Often, these cells are excluded by setting a minimum threshold on the number of UMIs per cell and/or genes detected. (2) Another aspect unique to droplet-based microfluidic devices is that the majority of the droplets (>90%) will not contain an actual cell[7,8]. Despite the absence of a cell, these "empty droplets" may contain low levels of background ambient RNA that was present in the cell solution[9]. An algorithm is needed to determine which droplets likely contained a real cell versus those that just contain ambient RNA[9]. Only droplets predicted to contain an actual cell are used in downstream analyses. (3) Doublets and multiplets occur when two or more cells are partitioned into a single droplet or well and will result in an artificial hybrid expression profile of each individual cell[2]. Several algorithms have been developed to identify potential doublets by combining expression profiles of randomly selected cells and then scoring each cell against the in silico doublets[10,11]. These tools can be used to flag problematic clusters that are created by droplets that contain cells from two different cell types. (4) Ambient RNA in the cell suspension can also be present in droplets containing a cell as well as empty droplets. These ambient transcripts will be counted along with a cell's native RNA and result in contamination of highly-expressed genes from other cell types. Tools such as decontX can be used to estimate contamination levels and deconvolute each cell into counts derived from native RNA and counts from contaminating ambient RNA[12]. (5) Perturbations during sample preparation can lead to biological artifacts. For example, cells that become stressed during tissue dissociation may express abnormally large proportions of mitochondrial genes in their transcriptome[13]. These cells may appear as a unique cluster in the scRNA-seq data even though they did not represent a unique cell population in the original tissue sample. If not taken into account, these factors can confound downstream analyses or produce erroneous findings. Therefore, performing comprehensive QC is a crucial step in scRNA-seq data analysis to ensure valid results.

While a large number of QC algorithms and software tools have been produced to address the specific challenges inherent in scRNA-seq data, these tools are implemented in different packages across various programming environments. In order to generate a comprehensive set of QC metrics, users need to separately download, install, and run each tool for each sample and independently assess the results[14]. Currently, there is a lack of standardized workflows that can streamline the process of generating QC metrics from different tools. While standalone quality control workflows such as those included in *Seurat* offer methods for the removal of poor-quality cells, it is usually focused on detection of droplets with abnormally low or high library size[15,16]. These workflows do not offer a breadth of options for QC, and do not contain methods for detection of doublets and ambient RNA contamination. In order to address these limitations, we have developed the SCTK-QC pipeline within the *singleCellTK* R package. The SCTK-QC pipeline is an extension of the *singleCellTK* package and designed as a standalone script which can be run on the command line and incorporated into standard cloud-based preprocessing pipelines. This pipeline can import single-cell RNA-seq data from a variety of preprocessing tools, run a multitude of different tools to generate comprehensive sets of QC metrics, visualize the results within detailed HTML reports, and export the results in an organized manner in various formats.

## Results

**Overview of SCTK-QC Pipeline**. The SCTK-QC pipeline is accessible through the *singleCellTK* package in R/Bioconductor. This pipeline assumes that the raw sequencing reads have been aligned, a correction for UMI and cell barcodes has been applied, and a count matrix containing genes and cells has been created by an upstream preprocessing tool such as CellRanger[7] or STARsolo[17]. For data generated with microfluidic devices, the first major step after UMI counting is to detect cell barcodes that represent droplets containing a true cell and exclude empty droplets that only contain ambient RNA[9]. We use the terms "Droplet" matrix to denote a count matrix that still contains empty droplets, "Cell" matrix to denote a count matrix of cells where empty droplets have been excluded but no other filtering has been performed, and "FilteredCell" matrix to indicate a count matrix where poor quality cells have also been excluded. The Droplet and Cell matrices have also been called "raw" and "filtered" matrices, respectively, by tools such as CellRanger. However, using the term "filtered" can be ambiguous as other forms of cell filtering can be applied beyond empty droplets (e.g., excluding poor-quality cells based on low number of UMIs). Additionally, even after excluding empty droplets and poor-quality cells, the matrix will still contain unnormalized counts, which is also commonly referred to as the "raw" count matrix. To eliminate ambiguity of these terms, we adopt the nomenclature of "Droplet", "Cell", and "FilteredCell" to describe the level of filtering on the dimensions of the count matrix while we prefer the terms "Raw", "Normalized", and "Scaled" to denote the level of processing for the counts within the matrix.

The major steps in the SCTK-QC pipeline include: (1) importing of the Droplet matrix, (2) detection and exclusion of empty droplets to create the Cell matrix, (3) calculation of a comprehensive set of QC metrics on the Cell matrix, (4) visualization of results in HTML format, and (5) exporting the data to formats used in downstream analysis workflows (Fig. 1). Note that several preprocessing tools automatically exclude empty droplets and create a Cell matrix. The SCTK-QC pipeline also has the ability to import a Cell matrix and start with the calculation of QC metrics in step 3 or import both the Droplet and Cell matrices and perform QC on each matrix independently. The single-cell data is stored within the pipeline as a *SingleCellExperiment* object[14]. Cell-level metrics generated by QC tools are stored in the *colData* slot alongside other imported cell-level annotations

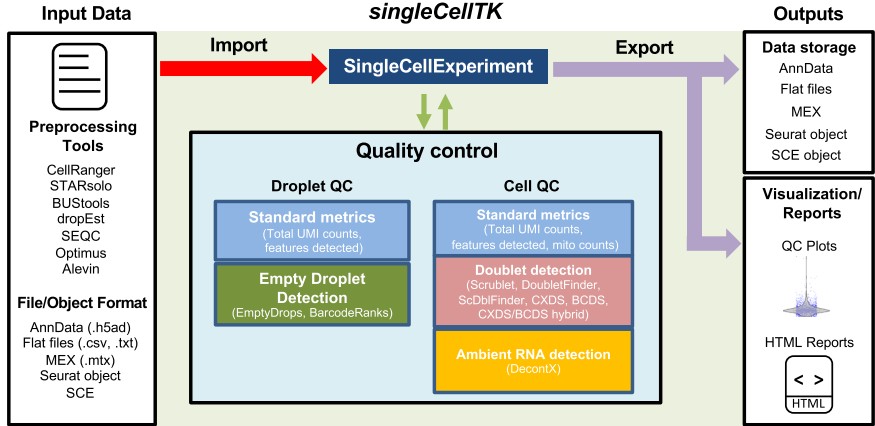

**Fig. 1 Overview of the SCTK-QC pipeline.** The SCTK-QC pipeline is developed in R and can import datasets generated from various preprocessing tools. The pipeline incorporates various software and tools to perform QC for Droplet and/or Cell matrices within each sample. Tools are included for calculation of standard metrics such as the number of Unique Molecular Identifier (UMIs) per cell, detection of empty droplets, prediction of doublets, and estimation of contamination from ambient RNA. The pipeline utilizes the *SingleCellExperiment* (SCE) R object to store assay data and the derived QC metrics. Data visualization and report generation can be subsequently performed on the imported dataset based on user specified parameters. All data can be exported to *Seurat* object, a Python *AnnData* object, or as Market Exchange Format (MEX) and .txt flat files to facilitate analysis in downstream workflows.

**Table 1 Functions available in the singleCellTK package and the SCTK-QC pipeline along with the corresponding wrapper functions.**

| SCTK QC modules | Methods | Goal | Packages integrated | Function |
|---|---|---|---|---|
| runDropletQC | runBarcodeRankDrops | Calculate barcode ranks | DropletUtils | barcodeRanks |
|  | runEmptyDrops | Detection of empty droplets | DropletUtils | emptyDrops |
|  | runPerCellQC | Compute general quality control metrics | scater | addPerCellQC |
| runCellQC | runPerCellQC | Compute general quality control metrics | scater | addPerCellQC |
|  | runScrublet | Doublet detection | Scrublet | scrub_doublets* |
|  | runScDblFinder | Doublet detection | scDblFinder | scDblFinder |
|  | runDoubletFinder | Doublet detection | DoubletFinder | doubletFinder_v3 |
|  | runCxds | Doublet detection | scds | cxds |
|  | runBcds | Doublet detection | scds | bcds |
|  | runCxdsBcdsHybrid | Doublet detection | scds | cxds_bcds_hybrid |
|  | runDecontX | Detect ambient RNA contamination | celda | decontX |

The diverse algorithms and their corresponding SCTK-QC wrapper functions that are used to generate quality control QC metrics in SCTK-QC pipeline. The asterisk denotes Python functions.

and corrected raw counts matrices created by any QC tool are stored in the *assays* slot. For reproducibility, the parameters and seeds used to run the functions within the pipeline will be also stored in the *metadata* slot. Overall, the pipeline supports importing data from 11 different preprocessing tools or file formats, empty droplet detection with 2 algorithms, generation of standard QC metrics, doublet detection using 6 algorithms, and estimation of ambient RNA with DecontX (Table 1). SCTK-QC integrates numerous tools across different programming environments such as R and Python. To streamline installation and minimize challenges with package dependencies, we have built Docker and Singularity images which are available through DockerHub (campbio/sctk_qc). To enable usage on the cloud, we have wrapped the SCTK-QC pipeline in Workflow Description Language (WDL) which can be used to QC samples on the Terra platform. The specific algorithms and tools used in each step of the pipeline are described in more detail below.

**Data import.** SCTK-QC can automatically import data from a variety of preprocessing tools and file formats. Supported preprocessing tools include CellRanger[7], BUStools[18], STARSolo[17], SEQC[19], Optimus[20], Alevin[21], and dropEST[22]. Generally, users will only need to specify the top-level directories for one or more samples and SCTK-QC will import and combine each sample into a single matrix (Supplementary Table 1). Alternatively, specific file formats such as Market Exchange Format (MEX) or a file containing comma-separated values (.csv) can be specified along with separate files for feature and cell annotation. By default, SCTK-QC will run QC analysis on both Droplet matrix and Cell matrix if both of them are provided. However, users can also choose to run QC only on the Droplet or only on the Cell matrix. The sample labels for each cell are stored in a variable called "sample" within the *colData* slot of the *SingleCellExperiment* object. Each QC algorithm will be applied to cells from each sample separately.

**Empty droplet detection.** Detection of empty droplets within the Droplet matrix is accomplished using the algorithms *barcodeRanks* and *EmptyDrops* from the *dropletUtils* package[9]. These algorithms are incorporated within the wrapper function *runDropletQC()*. *barcodeRanks* ranks all barcodes within the Droplet matrix based on total UMI counts per barcode. The knee, and inflection points are computed from the log-log plot of the rank against the total counts. Under the assumption that cells will have a higher number of total UMI counts than empty droplets, barcodes with total counts under the knee or inflection points are flagged as empty droplets. Rank, total counts, knee and inflection point are all outputted from the algorithm and stored within the *SingleCellExperiment* object. In contrast, *emptyDrops* differentiates between empty

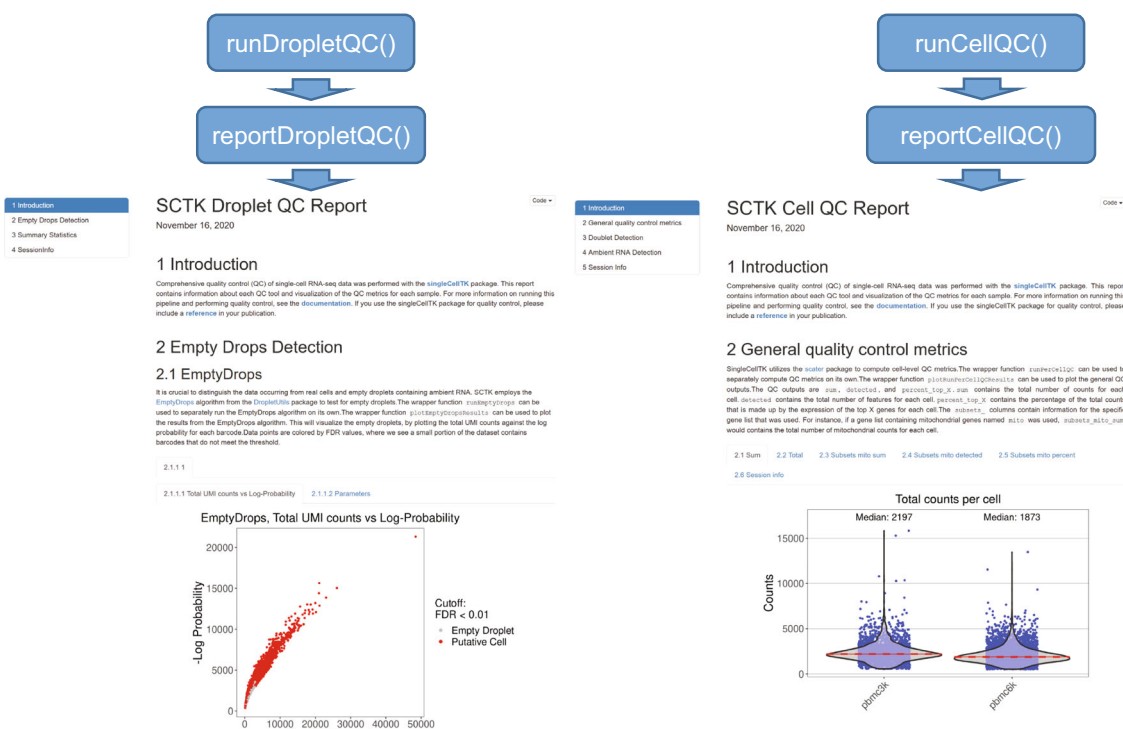

**Fig. 2 Generation of HTML reports for visualization and assessment of QC metrics.** The functions *reportDropletQC() and reportCellQC()* generate the extensive HTML reports to display data generated by the various QC tools applied by the functions *runDropletQC()* and *runCellQC()*, respectively. The *reportDropletQC()* report contains figures visualizing identified empty droplets. The *reportCellQC()* report contains visualizations of total read counts, total genes detected, doublet scores, doublet calls, percentages of ambient RNA detected, and cell clusters identified by decontX. These reports are run automatically by the SCTK-QC pipeline. Examples of *reportDropletQC()* (on the left*)* and *reportCellQC()* (on the right) reports are shown.

droplets containing ambient RNA from true cells by employing a probabilistic model that assumes a "pool" of ambient RNA from the Droplet matrix randomly contaminates each droplet. This algorithm identifies droplets containing true cells by comparing each droplet to a pool of low-count droplets that likely only contains ambient RNA. Metrics generated from this model include the total UMI counts per barcode, the log-probability, Monte Carlo *p*-value, and the *q*-value for a droplet containing a real cell, and a value signaling whether increasing the number of iterations within the algorithm will increase the likelihood of identifying a lower *p*-value. The *SingleCellExperiment* object containing the Droplet matrix can be automatically filtered on either the *barcodeRanks* or *emptyDrops* output to create a new *SingleCellExperiment* object containing the Cell matrix if this matrix was not originally supplied as input to the SCTK-QC pipeline.

**Generation of QC metrics**. Wrapper functions for each QC algorithm or tool are included in SCTK-QC. Additionally, the wrapper function *runCellQC()* is capable of executing these algorithms all at once within SCTK-QC. *runCellQC()* applies algorithms available from the *scater*[23] package to the Cell matrix to compute standard metrics. This includes the total UMI counts per cell, total number of features detected per cell, and the percentage of library size occupied by the most highly expressed genes in each cell. Users may also supply any gene set of their choice to calculate the aggregate expression of the gene set per cell. As a specific use case, a list of mitochondrial genes may be supplied to *runCellQC()* to compute the mitochondrial gene expression per cell. Mitochondrial gene sets for mouse and human in Gene Symbol, Ensembl and Entrez formats are stored in the package and can be supplied to the SCTK-QC pipeline by setting "-M" parameter. The *runCellQC()* function employs the following algorithms for doublet identification in the Cell matrix:

*Scrublet*[10], *scDblFinder*[24], *DoubletFinder*[11], and the *cxds*, *bcds*, and *cxds_bcds_hybrid* models from *SCDS*[25] package. All of these algorithms output a doublet score and derive a threshold to make a call as to whether each cell is a doublet or a singlet. Running multiple algorithms allows users to set their own criteria for flagging potential doublets that works best for their dataset. If users do not have a specific preference for a doublet detection algorithm, we recommend adopting a "consensus" approach and consider cells identified as doublets by multiple algorithms to most likely be doublets. Finally, the *runCellQC()* function runs decontX[12] to detect ambient RNA contamination for each cell within the Cell matrix. The percentage of estimated contamination is stored within the *colData* and the decontaminated count matrix is stored as an assay in the *SingleCellExperiment*, which can be optionally used in downstream analysis. After completion of *runCellQC()*, users can use any combination of these QC metrics to filter the Cell matrix and create a FilteredCell matrix for use in downstream analyses. For all of these above QC metrics, users may determine their own outlier cutoff by applying the wrapper function *detectCellOutlier()* which determines outliers based on median absolute deviation.

**Generation of comprehensive QC HTML reports**. Rmarkdown documents can be used to create dynamic HTML or PDF reports useful for systematic display and evaluation of data[26]. We include the functions *reportDropletQC()*, *reportCellQC()*, and *reportQC-Tool()*, which make use of algorithm-specific Rmarkdown document templates to generate HTML reports with the visualizations of QC metrics from all algorithms (Fig. 2). *reportDropletQC()* generates a report including a scatterplot annotating all empty droplets flagged by the EmptyDrops algorithm as well as a curve visualizing the knee and inflection points identified by Barco-deRanks. For each set of doublet detection algorithm executed,

*reportCellQC()* generates a report which visualizes the doublet score and call through violin plots, density plots, and dimensionality reduction plots. These plots are also created to visualize the contamination percentage of ambient RNAs computed by decontX if the algorithm has been applied to the data. Additionally, both reports include a summary table detailing the outputted quality control metrics for all algorithms run.

**Export to common data structures**. Different software packages utilize varying data containers to store and retrieve scRNA-seq data[27]. To facilitate downstream analysis in multiple platforms, the SCTK-QC pipeline provides several functions to export the data in one or more data structures or file formats. The *exportSCEtoFlatFile()* function writes assays to MEX files and the *colData*, *rowData*, *reducedDims* slots into tab-delimited flat files. The metadata is exported as a list in an RDS file. All exported files can be optionally saved in a gzipped format. The *exportSCEtoAnnData()* function exports the data into a Python annotated data matrix (*AnnData*)[28] object. The function stores assay, rowData, colData and reducedDims slots into *X*, *var*, *obs*, and *obsm* groups of the *AnnData* object, respectively. The *AnnData* object can be written into a.h5ad file format and can subsequently be compressed in a "gzip" or "lzf" format. The convertSCEToSeurat function exports the data into a *Seurat*[15,16] object. The function stores assay, colData and reducedDims slots into assays, meta.data and reductions groups of the *Seurat* object. These functions can be run by setting the "-F" or "--outputFormat" parameter in the SCTK-QC pipeline.

**R/Shiny user interface**. Shiny is a R package developed for building interactive web applications. The *singleCellTK* package incorporates Shiny Graphical User Interface (GUI) for the interactive analysis of single-cell data. Users are able to access the user interface by simply executing the *singleCellTK*() function in the R console. Upon loading the data, QC algorithms to be run are chosen on the "Data QC & Filtering" page by selecting checkboxes in the user interface (Fig. 3). Upon the completion of the algorithms, QC plots will appear within tabs for each of the algorithms selected. The "Filtering" tab can be used to set criteria for filtering. After QC, users are able to interactively perform other downstream analyses such as batch correction, feature selection, dimensionality reduction, clustering, differential expression, and pathway analysis. Upon completion, the data can be exported as a RDS, Python *AnnData*, or a tab-delimited flat file. Benchmarking of the Shiny GUI was conducted on a Linux machine using example datasets of 34k and 68k PBMCs. Due to one of the doublet detection algorithms, *DoubletFinder*, requiring large amounts of memory, benchmarking was performed with and without executing *DoubletFinder*. When *DoubletFinder* was applied, the 34k and 68k PBMC dataset required 18.5GB and 4200 seconds, and 38.5GB and 6820 seconds, respectively. When *DoubletFinder* was removed from the list of algorithms executed, the 34k and 68k PBMC dataset required 6GB and 2350 seconds, and 7GB and 3960 seconds, respectively (Supplementary Table 2). Therefore, all of the QC tools with the possible exception of *DoubletFinder* can be applied on larger datasets using the Shiny GUI.

**Comparison to other tools**. Several other tools that can perform single-cell RNA sequencing data analysis and quality control have been created. While many packages only support input data stored in structured format (*SingleCellExperiment* object, *Seurat*[15,16] object or count matrix stored in csv/txt/mtx file), SCTK-QC also accepts data generated from different preprocessing tools and .h5ad files. Although other packages can generate

general QC metrics including number of UMIs and features detected per cell, SCTK-QC includes comprehensive QC analysis including empty droplet detection, doublet detection and ambient RNA correction (Table 2). Furthermore, no other software package currently runs multiple doublet detection methods and allows users to easily compare results. SCTK-QC also visualizes QC metrics in standardized html reports and stores results in several data formats, which facilitates downstream analysis in different analysis workflows. Currently, SCTK-QC does not support RSEM as an input format and it does not support other Python objects such as *pickle* and *joblib* as these are not commonly used.

**Application of SCTK-QC pipeline to PBMC datasets**. To demonstrate the utility of SCTK-QC, we apply the pipeline to the 10x Genomics 1K healthy donor Peripheral Blood Mononuclear Cell (PBMC) dataset generated with v2 or v3 Chromium chemistries. Each dataset was processed with two different versions of Gencode GTF files (Gencode v27 and Gencode v34). The resulting four count matrices (Gencode v27 PBMC 1K v2, Gencode v27 PBMC 1K v3, Gencode v34 PBMC 1K v2, Gencode v34 PBMC 1K v3) were then processed by the SCTK-QC pipeline. Specifically, the pipeline used the *importCellRangerV2()* and *importCellRangerV3()* function by setting the "-cellRangerDirs" as the path of input data, and the "-dataType" parameter as "Cell" and the *runCellQC()* function was called to generate the QC metrics. All of the QC metrics are summarized for each of the four samples in Table 3. The distributions for some of the general QC metrics and the decontX decontamination scores are displayed in violin plots. (Fig. 4a). As expected, the median counts and features detected in the alignments from v3 chemistry PBMC datasets were almost double than those detected from v2 chemistry indicating the higher capture sensitivity of the 10x v3 chemistry. No significant difference was observed in the total read counts ($p = 0.93$; *t*-test) and the number of features detected per cell between the PBMC datasets aligned to different versions of Gencode references (*p-value*: 0.69; *t*-test). The predicted doublet rate of each dataset varied among different doublet detection methods. With the exception of DoubletFinder and scDblFinder, all other methods consistently predicted higher doublets rate for v3 chemistry dataset than those for v2 chemistry dataset. Finally, lower decontX contamination scores suggest improved processing for the samples profiled with the v3 chemistry.

We additionally applied the SCTK-QC pipeline to two replicate human PBMC datasets previously generated with the SMART-Seq2 protocol, available from the Human Cell Atlas Single Cell Portal[20,29]. The QC metrics for each replicate are summarized in Table 3. The distributions for the library size, total features detected and the decontX decontamination scores per cell are displayed in violin plots (Fig. 4b). The overall library size, as well as the number of features detected per cell was significantly higher in the SMART-Seq2 datasets than the 10X datasets (Library size: $p < 2E\text{-}16$, Number of features: $p < 2E\text{-}16$; *t*-test). In all doublet detection algorithms, the doublet rate was higher in the 10X protocol compared to the SMART-Seq2 protocol. The mean ambient RNA contamination level was 1.8 times higher on average in the 10X droplet-based datasets compared to the two SMART-Seq2 replicates.

## Discussion

The wide applicability of single-cell approaches has led to the development of novel computational tools that allow for clustering and identification of new cell types and trajectory inference of cell populations in development. Despite the improvements of scRNA-seq platforms and protocols, low-quality cells and

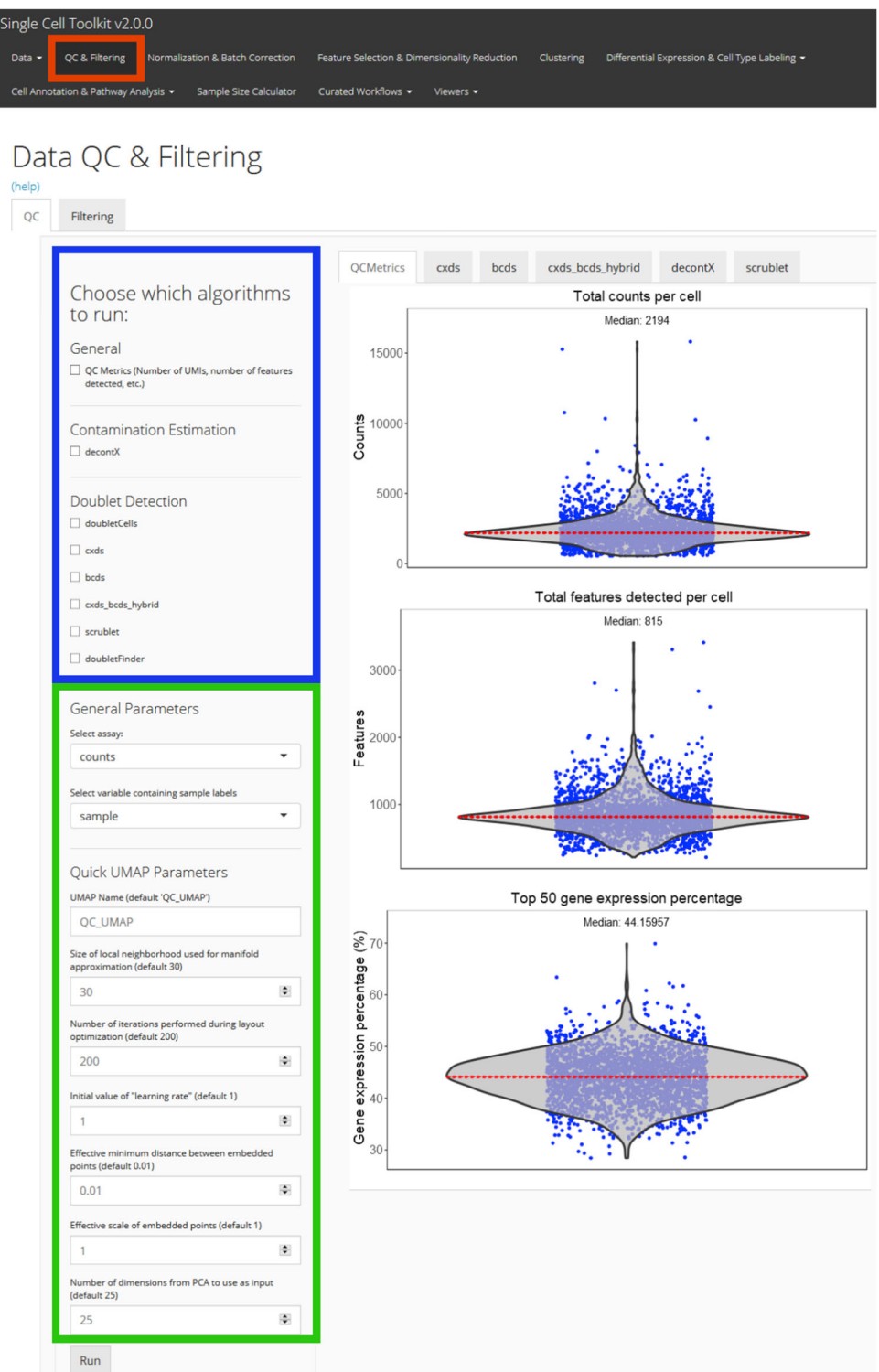

**Fig. 3 Interactive QC of single cell data using a Graphical User Interface (GUI).** An R/Shiny GUI can be used to interactively run QC algorithms in the *singleCellTK* package. A screenshot of the "Data QC & Filtering" tab from the interactive GUI is shown. After importing the data, quality control is performed within the "QC & Filtering" tab (red) of the user interface. QC algorithms are chosen from a list (blue), while specific parameters may be specified as well (green). Plots displaying metrics generated by each QC tool will appear to the right in a tab.

technical artifacts such as empty droplets, doublets, and ambient RNA still remain present to some degree in most datasets. Thus, rigorous QC measures are needed to evaluate the quality of individual experiments. The SCTK-QC pipeline streamlines and standardizes the generation and visualization of metrics important for assessing data quality. Previous tools like *FastQC* and

*RSeQC*[30,31] have enabled extensive quality assessment and visualizing of FASTQ and aligned BAM files. Similarly, the SCTK-QC pipeline enables comprehensive generation and visualization of QC metrics for the initial count matrix by integrating several algorithms and tools into a common, easy-to-run framework. Importantly, SCTK-QC pipeline provides a

**Table 2 Comparison of features in the SCTK-QC pipeline with other single-cell analysis toolkits.**

| | SCTK | PIVOT | Seurat | ascend | scRNABatchQC | Adobo | SCONE | SCHNAPPs | iS-CellR | Ganatum | ASAP browser |
|---|---|---|---|---|---|---|---|---|---|---|---|
| *Input format* | | | | | | | | | | | |
| 10x CellRanger | ✓ | | ✓ | | ✓ | | | | ✓ | | |
| SCE Object | ✓ | | | | ✓ | | ✓ | ✓ | ✓ | | |
| Seurat Object | ✓ | ✓ | | | | | | | | | |
| AnnData | ✓ | | | | | | | | | | ✓ |
| LOOM | | | | | | | | | | | |
| BUStools | ✓ | | | | | | | | | | |
| SEQC | ✓ | | | | | | | | | | |
| STARSolo | ✓ | | | | | | | | | | |
| Optimus | ✓ | | | | | | | | | | |
| DropEst | ✓ | | | | | | | | | | |
| CSV, TXT, and MTX | ✓ | ✓ | ✓ | ✓ | ✓ | ✓ | | ✓ | ✓ | ✓ | ✓ |
| RSEM | | | | | | | ✓ | | | | |
| *Ambient droplets detection* | ✓ | | | | | | | | | | |
| *General QC Metrics* | | | | | | | | | | | |
| Total counts | ✓ | ✓ | ✓ | ✓ | ✓ | ✓ | ✓ | ✓ | ✓ | ✓ | ✓ |
| Number of features detected | ✓ | ✓ | ✓ | ✓ | ✓ | ✓ | ✓ | ✓ | ✓ | ✓ | ✓ |
| Gene set count (e.g mitochondrial) | ✓ | ✓ | ✓ | ✓ | ✓ | ✓ | | | ✓ | | ✓ |
| *Doublet detection* | | | | | | | | | | | |
| scDblFinder | ✓ | | | | | | | | | | |
| Scrublet | ✓ | | | | | | | | | | |
| doubletFinder | ✓ | | | | | | | | | | |
| cxds | ✓ | | | | | | | | | | |
| bcds | ✓ | | | | | | | | | | |
| cxds/bcds hybrid | ✓ | | | | | | | | | | |
| *Shiny App/interactive* | ✓ | ✓ | | | | | ✓ | ✓ | ✓ | ✓ | ✓ |
| *docker* | ✓ | ✓ | | | | | | | ✓ | | ✓ |
| *HTML Report* | ✓ | | | ✓ | ✓ | | | | | | ✓ |
| *Output format* | | | | | | | | | | | |
| RDS | ✓ | | | ✓ | ✓ | | | ✓ | | | |
| AnnData | ✓ | | | | | | | | | | |
| hdf5 | ✓ | | | | | | | | | | ✓ |
| .txt Flatfile | ✓ | | | | | | | ✓ | | | |
| pickle | | | | | | ✓ | | | | | |
| joblib | | | | | | ✓ | | | | | |

SCTK-QC pipeline supports various types of input, full scRNA-seq quality control pipeline and supports common data structures for data storage.

framework with standardized data structures for computing and storing QC metrics. This modular architecture of SCTK-QC will allow for easy integration of new tools as they are made available in the future. SCTK-QC is able to generate HTML reports with publication-ready figures and contains a GUI for interactive QC of single-cell data. These features will enable users without in-depth programming backgrounds to run these tools and perform QC on their data. Finally, the SCTK-QC pipeline can export both R and Python-compatible data structures enabling easy integration with other popular analysis frameworks such as *Seurat*[15,16] and *Scanpy*[28]. While packages such *Seurat* or *Scanpy* offer the ability to calculate some basic QC metrics and remove poor quality cells, they do not have a comprehensive workflow that includes multiple algorithms for detection of doublets or estimation of contamination from ambient RNA. Additionally, no other package includes the ability to produce HTML reports for easier assessment of quality metrics.

## Methods

**Accessibility**. The SCTK-QC pipeline is executable on the R console, Rstudio or on the Unix command-line with an Rscript command. The *singleCellTK* package and quality control pipeline is open sourced through GitHub (https://github.com/compbiomed/singleCellTK) and the Bioconductor repository. Additionally, we have included scripts to set up the Conda or Python virtual environments that meet all cross-platform dependency requirements for convenient portability of the pipeline between operating systems. To encourage reproducibility and make the computing environment independent, the *singleCellTK* package and SCTK-QC pipeline is included in Docker image[32] (https://hub.docker.com/r/campbio/sctk_qc). All dependencies of the *singleCellTK* package are included in the Docker image and the quality control pipeline can be executed with a single docker run. Users can specify parameters used for each QC function by providing a YAML file to the pipeline with the argument "-y" or "--yamlFile". We have created several vignettes and in-depth walkthroughs for installation and analysis workflows which are available on the GitHub repository and at https://www.camplab.net/sctk. To enable QC analysis on cloud, SCTK-QC is wrapped in WDL, which can be deployed on Terra platform (https://github.com/htan-pipelines/scrna-seq-pipeline/tree/master).

**Quality control of PBMC datasets**. *10X Genomics*. The raw reads in the FASTQ format were downloaded from the 10x Genomics Dataset portal and the human reference genome sequence GRCh38 release versions 27 and 34 in the FASTQ and GTF formats from the GENCODE website. The "mkref" command in *CellRanger* v3.1.0 was used to build separate custom references for Gencode GRCh38 v27 and v34. Droplet and Cell matrices for both PBMC 1k v2 and v3 samples were then obtained by aligning the raw reads to the reference genomes using *CellRanger* v3.1.0 running *bcl2fastq* v2.20.

*SMART-Seq2*. The raw read count data from two PBMC SMART-Seq2 replicates available from the "counts.read.txt.gz" file located at https://singlecell.broadinstitute.org/single_cell/study/SCP424/single-cell-comparison-pbmc-data was processed through the SCTK-QC pipeline. A *SingleCellExperiment* object was constructed from the available cell names, gene names, and the metadata, labeled "cells.read.new.txt", "genes.read.txt", and "meta.counts.new.txt", respectively. Both 10X and SMART-Seq2 dataset size is summarized in Table 3.

**Table 3 Summary of QC metrics for each PBMC sample. A total of six PBMC datasets were analyzed with the SCTK-QC pipeline.**

| | GENCODE GRCh38 v27 | | GENCODE GRCh38 v34 | | SMART-Seq2 | |
|---|---|---|---|---|---|---|
| | PBMC1k V2 | PBMC1k V3 | PBMC1k V2 | PBMC 1k V3 | Replicate 1 | Replicate 2 |
| Total number of genes detected | 58,347 | 60,669 | 58,347 | 60,669 | 33,694 | 33,694 |
| Number of droplets, Droplet matrix | 737,280 | 6,794,880 | 737,280 | 6,794,880 | NA | NA |
| Number of Cells, Cell matrix | 995 | 1223 | 996 | 1222 | 311 | 273 |
| Mean counts | 3559 | 7576 | 3553 | 7576 | 390,058 | 292,971 |
| Median counts | 3374 | 6637 | 3375 | 6640 | 388,420 | 290,819 |
| Mean features detected | 1133 | 2088 | 1140 | 2104 | 2436 | 2795 |
| Median features detected | 1106 | 1957 | 1109 | 1978 | 2406 | 2632 |
| Scrublet, Number of doublets | 12 | 16 | 12 | 18 | 0 | 3 |
| Scrublet, Percentage of doublets | 1.21 | 1.31 | 1.2 | 1.47 | 0 | 1.1 |
| ScDblFinder, Number of doublets | 13 | 16 | 14 | 20 | 3 | 13 |
| ScDblFinder, Percentage of doublets | 1.31 | 1.31 | 1.41 | 1.64 | 0.97 | 4.76 |
| DoubletFinder, Number of doublets, Resolution 1.5 | 75 | 92 | 75 | 92 | 23 | 20 |
| DoubletFinder, Percentage of doublets, Resolution 1.5 | 7.54 | 7.52 | 7.53 | 7.53 | 7.4 | 7.33 |
| CXDS—Number of doublets | 51 | 195 | 53 | 183 | 19 | 4 |
| CXDS—Percentage of doublets | 5.13 | 15.9 | 5.32 | 15 | 6.11 | 1.47 |
| BCDS—Number of doublets | 91 | 91 | 69 | 71 | 17 | 8 |
| BCDS—Percentage of doublets | 9.15 | 7.44 | 6.93 | 5.81 | 5.47 | 2.93 |
| SCDS Hybrid—Number of doublets | 65 | 119 | 77 | 90 | 20 | 13 |
| SCDS Hybrid—Percentage of doublets | 6.53 | 9.73 | 7.73 | 7.36 | 6.43 | 4.76 |
| DecontX—Mean contamination percentage | 5.4 | 3.7 | 5.8 | 3.0 | 2.1 | 2.9 |
| DecontX—Median contamination percentage | 1.7 | 0.9 | 1.8 | 0.7 | 0.7 | 1.3 |

A total of six PBMC datasets were analyzed with the SCTK-QC pipeline. Two GENCODE PBMC 1k datasets of differing 10x Chemistry were taken from GENCODE v27 and v34, resulting in a total of four datasets. Additionally, two SMART-Seq2 datasets from PBMC replicates were also taken. A per-sample summary table is automatically generated by the pipeline.

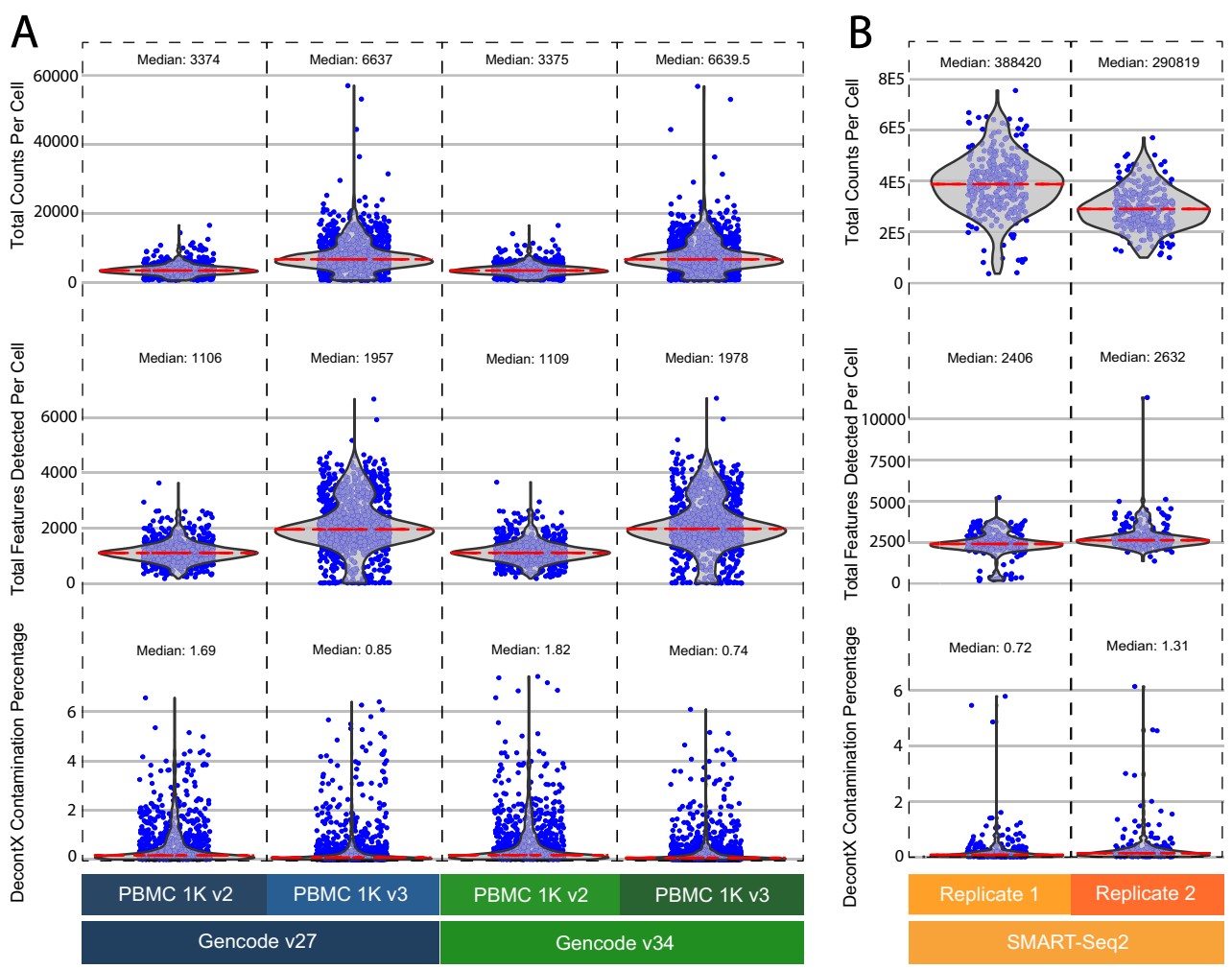

**Fig. 4 Application of SCTK-QC to PBMC datasets. A** QC metrics were generated by the SCTK-QC pipeline for 1K healthy donor Peripheral Blood Mononuclear Cell (PBMC) datasets from 10X Genomics. Violin plots generated by the pipeline demonstrate higher capture sensitivity of the 10x v3 Chromium chemistry. Furthermore, lower ambient RNA contamination was observed in the samples run with v3 chemistry compared to samples profiled with the v2 chemistry. **B** The SCTK-QC pipeline was applied similarly on a PBMC dataset generated by SMART-Seq2. A higher number of features were detected per cell in the SMART-Seq2 datasets compared to either of the 10X Genomics datasets.

Quality control on count matrices was conducted with SCTK-QC under default parameters for all QC algorithms.

**Reporting summary**. Further information on research design is available in the Nature Research Reporting Summary linked to this article.

## Data availability

Both V2 and V3 chemistry PBMC data used in this study is available on the 10X Genomics website database [https://www.10xgenomics.com/resources/datasets/1-k-pbmcs-from-a-healthy-donor-v-2-chemistry-3-standard-3-0-0, (V2), https://www.10xgenomics.com/resources/datasets/1-k-pbmcs-from-a-healthy-donor-v-3-chemistry-3-standard-3-0-0 (V3)]. The SMART-seq2 PBMC data used in this study is available in the Single Cell Portal [https://singlecell.broadinstitute.org/single_cell/study/SCP424/single-cell-comparison-pbmc-data]. The processed PBMC data is available on GitHub at https://github.com/campbio/Manuscripts/tree/master/Hong_SCTK-QC under the "Data" folder.

## Code availability

All code used to generate QC analysis results and figures are available on GitHub at https://github.com/campbio/Manuscripts/tree/master/Hong_SCTK-QC under the "Scripts" folder or under the https://doi.org/10.5281/zenodo.6256957.

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

## Acknowledgements

This work was funded by the National Library of Medicine (NLM) R01LM013154-01 (J.D.C. and M.Y.), the National Cancer Institute (NCI) Informatics Technology for Cancer Research (ITCR) 1U01 CA220413-01 (W.E.J. and J.D.C.), the National Institute of General Medical Sciences of the National Institutes of Health (NIGMS) T32GM100842 (Y.K., S.B., and A.L.), 5R01GM127430 (W.E.J.), and the NCI Human Tumor Atlas Network (HTAN) Pre-Cancer Atlas (PCA) 1U2CCA233238 (J.D.C.).

## Author contributions

Software design (R.H., Y.K., S.B., A.L., Y.W., V.A., X.C., I.S., Z.W., S.A., F.J., M.Y., W.E.J., and J.D.C.); data analyses (R.H., Y.K., S.B., A.L., and J.D.C.); writing of the manuscript (R.H., Y.K., S.B., A.L., and J.D.C.); editing of the manuscript (R.H., Y.K., S.B., A.L., Z.W., S.A., W.E.J., and J.D.C.)

## Competing interests

The authors declare no competing interests.
