## [Peer Review File · Nature Communications]

Reviewers' Comments:

Reviewer #1:

Remarks to the Author:

I like the idea of generating standardized and reproducible workflows which ends in easy to use for non-bioinformaticians and interactive visualisations. The goal of nature communications is to present "represent important advances of significance to specialists within each field". I think this tool is nice to make analysis 'easier' for some people, but there is not enough novelty to call it an important advancement. In this case the user has still many decisions to make and best practices/benchmarks are lacking. On top, the installation and user manuals/vignettes are not working as they should be.

Major comments

The pipeline is demonstrated with 10X data. Demonstrations with full length single cell data are lacking (they might additional QC steps, eg. gene body coverage, ...). I think the data types included are too limiting. What with the current advancements with CITE-seq data and for instance scATAC-seq?

For that kind of tools, an explanatory manual is key. Seurat has easy to follow vignettes for all types of analysis, which makes it easy to implement. I miss a link to the available vignettes for this package and instructions on the github page. I found the vignettes, but the descriptions are limited. What cut-offs should you use? Which tools are the best? When I go to Bioconductor and click on links to see how it works, these are not working ("page not found" error pages pop up instead). I tried to installed both the Bioconductor library and the github library. Bioconductor installation was working, but the first function that I needed "importCellRanger" was not found. Then I tried to install the github library but that failed "Error in parse(outFile) : /private/var/folders/q2/pqzrkbmj5ngfzlw_p77rwntm0000gn/T/RtmpKAuXY3/R.INSTALL7bdb66a30133/singleCellTK/R/readSingleCellMatrix:75:13: unexpected symbol".

They mention that information is available at: <https://github.com/campbio/Manuscripts/SCTK-QC> But this webpage is not found as well.

In the end, I was not able to test this package with my own data in R.

Minor comments:

Line 57: "majority of droplets". Can you specific numbers?

They mention the lack of standardized workflows? What about Seurat, they also do some kind of QC?

In the processing tools, Alevin compatibility would be great, since this is used more and more.

What with export to Seurat or Monocle proof-objects?

Shiny tools are nice, but do require RAM/ CPU/... what are the requirements for your tools scaled to number of cells?

Comparison to other tools: Seurat itself might not be able to import/export, but many tools are foreseen in Seurat-disk.

Reviewer #2:

Remarks to the Author:

This manuscript describes the SCTK-QC pipeline, an R package able to run and process outputs from a number of single-cell analysis bioinformatics tools. In addition to data standardisation and visualisation, the package comes with tools to filter and prepare single cell transcriptomic data for downstream analysis.

The major claims of the paper are that this tool brings together diverse packages into a single consistent interface, suitable for users who may not be familiar with the complexities of installation and execution of varied command line bioinformatics tools. As the number and size of single cell transcriptomics studies is expanding rapidly at the moment, I agree with the authors that this is a worthwhile aim and of importance for the research community. Data quality control and filtering is important for any research project, but it is especially critical for single cell data. By making such analysis more accessible and user-friendly, more research projects are likely to adhere to best-practices for their analysis.

I thought that the manuscript text was very well written - clear and accessible for newcomers to the field. I have almost no comments on the text itself, I thought it was excellent.

My main concern is the difference between the SingleCellTK R package and the SCTK-QC pipeline. I understand that the paper is about the -QC pipeline and not the R package, but I think that the manuscript would benefit from clearer disambiguation between the two.

I tried to look into the code used for the paper, but the supplied URL (<https://github.com/campbio/Manuscripts/SCTK-QC>) gave a 404 not accessible error. I guess that this repository is currently set to private.

I am not a very experienced user of R so consider myself well positioned to try out the author's claim of user-friendliness of their package. I attempted to install and run the software myself, but found the tool documentation very difficult to navigate. It is quite spread out (GitHub readme, a GitHub repo wiki, the singleCellTK homepage and the Docker Hub readme - all seem to contain different information). It would be good if one was chosen and the others cut down to just point to that canonical link.

The website navigation for the singleCellTK website (<https://www.sctk.science>) also seems to be broken - some pages do not exist (under Command Line Analysis dropdown), the navigation changes depending on what page you are on (Command Line Analysis > Batch Correction) and a number of nav links only work from the homepage (following from other pages, articles/ keeps getting added to the URL). Some content I could only find by taking specific paths through the website by trial and error, such as the Shiny UI Reference. Some documentation also seemed out of sync between GitHub and the website (eg. the difficult to find https://github.com/complbiomed/singleCellTK/blob/master/exec/SCTK_runQC_Documentation2.md and https://www.sctk.science/articles/run_qc.html).

The documentation itself is generally extensive but again suffers from a lack of clarity between the different subtools. It would be good to have an immediate introduction to the different entry points for an end user at the start of the docs - R session / command line interface / Docker / Shiny. In fact I couldn't find any documentation about running the Shiny UI anywhere except for the mention of the singleCellTK() function in the manuscript. It would be great if the Shiny app could also be bundled into a Docker image to make it faster and easier to run.

I tried following the installation instructions for the singleCellTK package in R on a relatively new system and although it has run smoothly so far it is incredibly slow (it's been going for nearly three hours now as I write this). As such I tried to use Docker image instead but the command given (`docker pull campbio/sctk_qc`) fails because it defaults to the :latest tag which does not exist. Running the latest tag (`docker pull campbio/sctk_qc:2.0.0`) does work. There is no Dockerfile for the image at Docker Hub or in GitHub so it wasn't entirely clear to me what the image contained. I tried to run the container in the hope that I could test the singleCellTK package within it but it failed with an error (Error in strsplit(opt[["preproc"]], ",") : non-character argument). As a last stab I tried the example docker command on an empty directory to see how far it would get but I again got a loading error (Error in match(x, table, nomatch = 0L) : 'match' requires vector arguments).

Given this plus the lack of the manuscript repo code to follow I have now given up trying to run the tool myself - either the command line tools, the Shiny app or the R functions. I recommend that the authors try to get a few people outside of their normal community to run the tool as I did

using only the available documentation, to try to iron out these problems. I suspect that most will be quite trivial to solve. Apologies if I missed something obvious.

In conclusion, I think that the manuscript is excellent and the aim and example outputs of the tool are superb. I think that greater clarity is required between the different components of this package, especially if this paper is about only part of the total singleCellTK suite. If the QC pipeline is to be successful in targeting computationally inexperienced users then the documentation needs quite a bit of work.

Philip Ewels

Reviewer #1 (Expertise: scRNAseq analysis):

I like the idea of generating standardized and reproducible workflows which ends in easy to use for non-bioinformaticians and interactive visualizations. The goal of nature communications is to present “represent important advances of significance to specialists within each field”. I think this tool is nice to make analysis ‘easier’ for some people, but there is not enough novelty to call it an important advancement. In this case the user has still many decisions to make and best practices/benchmarks are lacking. On top, the installation and user manuals/vignettes are not working as they should be.

Response 1.0. We appreciate the reviews note about the importance of standardized and reproducible workflows that can be used by non-computational researchers. In the following responses, we will describe how we have improved the documentation and vignettes. To increase the impact of this software, we have also added approaches for automated detection of outliers for each QC metric to aid the users in decision making.

Major comments

The pipeline is demonstrated with 10X data. Demonstrations with full length single cell data are lacking (they might additional QC steps, eg. gene body coverage, ...). I think the data types included are too limiting. What with the current advancements with CITE-seq data and for instance scATAC-seq?

Response 1.1. We agree that full-length scRNA-seq protocols offer advantages that 3’ and is still used widely enough to be considered by our pipeline. To further demonstrate the utility of SCTK-QC, we have applied SCTK-QC to a SMART-seq2 PBMC dataset from the Human Cell Atlas. We have included the following text describing our analysis on the SMART-seq2 dataset:

“We additionally applied the SCTK-QC pipeline to two replicate human PBMC datasets previously generated with the SMART-Seq2 protocol, available from the Human Cell Atlas Single Cell Portal. The QC metrics for each replicate are summarized in Table 3. The distributions for the library size, total features detected and the decontX decontamination scores per cell are displayed in violin plots. (**Figure 4b**). The overall library size, as well as the number of features detected per cell was significantly higher in the SMART-Seq2 datasets than the 10X datasets (Library size: $p < 2E-16$, Number of features: $p < 2E-16$; t-test). In all doublet detection algorithms, the doublet rate was higher in the 10X protocol compared to the SMART-Seq2 protocol. The mean ambient RNA contamination level was 1.8 times higher on average in the 10X droplet-based datasets compared to the two SMART-Seq2 replicates.”

With respect to CITE-seq and scATAC-seq data, we agree that these are other important advancements in the field. While SCTK-QC is currently the most comprehensive QC pipeline for scRNA-seq data, the SCTK-QC could be expanded to include quality control for these other data types. We are currently in the process of making another R/Bioconductor package called “sclmport” which will automatically import and store multi-modal data from 10X and other

platforms in Bioconductor Experiment-like objects. We will then expand the *singleCellTK* package to include workflows for QC'ing these data types in the future.

For that kind of tools, an explanatory manual is key. Seurat has easy to follow vignettes for all types of analysis, which makes it easy to implement. I miss a link to the available vignettes for this package and instructions on the github page. I found the vignettes, but the descriptions are limited.

Response 1.2. We apologize for the lack of clarity in our previous vignettes and the troubles finding the latest documentation. We have overhauled the documentation for running SCTK-QC in the R console (https://camplab.net/sctk/v2.4.1/articles/articles/cnsl_cellqc.html), in the R/Shiny app (https://camplab.net/sctk/v2.4.1/articles/articles/ui_qc.html), and on the command-line using the Docker image (https://camplab.net/sctk/v2.4.1/articles/articles/cmd_qc.html). Other tutorials can be found at <https://camplab.net/sctk>. The previous domain (sctk.science) will be phased out.

What cut-offs should you use? Which tools are the best?

Response 1.3. For the general QC metrics such as number of UMIs or number of genes detected, we have incorporated functionality to detect outliers within each distribution. These thresholds can be used as a guide to cut off the “tail” ends of these distributions. This update is described in the text, under “3. Generation of QC metrics”:

“For all of these above QC metrics, users may determine their own outlier cutoff by applying the wrapper function *detectCellOutlier()* which determines outliers based on median absolute deviation.”

With respect to doublet detection algorithms, each tool provides a default cutoff and “call” as to whether each cell is a doublet or not. Therefore, we do not provide additional cutoffs for these tools. In general, it can be difficult to assess which doublet detection tool is best and different tools may work better in different situations. Therefore, we suggest a “consensus” approach where cells that are flagged as doublets by multiple algorithms should be considered the most likely to be doublets. These changes have been described in the following text, under “3. Generation of QC metrics”:

“If users do not have a specific preference for a doublet detection algorithm, we recommend adopting a “consensus” approach and consider cells identified as doublets by multiple algorithms to most likely be doublets.”

When I go to Bioconductor and click on links to see how it works, these are not working (“page not found” error pages pop up instead).

Response 1.4. As mentioned previously, the updated documentation has been moved to <https://camplab.net/sctk>. The links to these documents on the Bioconductor version have been

updated as well.

I tried to install both the Bioconductor library and the github library. Bioconductor installation was working, but the first function that I needed “importCellRanger” was not found. Then I tried to install the github library but that failed “Error in parse(outFile) :

```
/private/var/folders/q2/pqzrkbmj5ngfzlw_p77rwntm0000gn/T/RtmpKAuXY3/R.INSTALL7bdb66a30133/singleCellTK/R/readSingleCellMatrix:75:13: unexpected symbol”.
```

Response 1.5. We apologize for these inconveniences as the *singleCellTK* package is still under active development. The previous version of the package in Bioconductor 3.12 contained an error in the function “importCellRanger”. The latest version of this package that was recently published in May in Bioconductor 3.13 had fixed this function. Therefore, re-installation of this package from the latest version of Bioconductor will contain all of the functions used in this manuscript. Additionally, the master branch on GitHub also had a temporary error which has been fixed. We have conducted additional tests on Windows, MacOS, and Linux to ensure that the package can be installed on multiple platforms.

They mention that information is available at: <https://github.com/campbio/Manuscripts/SCTK-QC> But this webpage is not found as well. In the end, I was not able to test this package with my own data in R.

Response 1.6. We apologize for this oversight. The example code used to create the data in Figure 4 and Table 3 of the manuscript is now available on the GitHub repo. https://github.com/campbio/Manuscripts/tree/master/Hong_SCTK-QC

Minor comments:

Line 57: “majority of droplets”. Can you specific numbers?

Response 1.7. The number of empty droplets can vary depending on the technology used and the concentration of cells in the suspension. Across the four 10X PBMC datasets, we observed 99.87-99.98% of barcodes were predicted to be an empty droplet by the emptyDrops package. For clarity, we have added a specific number for the percentage of droplets that does not contain a cell. This number has been reported in a previous study by Ni et al.

“Another aspect unique to droplet-based microfluidic devices is that the majority of the droplets (>90%) will not contain an actual cell¹.”

They mention the lack of standardized workflows? What about Seurat, they also do some kind of QC?

Response 1.8. The Seurat package does provide the capability to calculate and filter on some QC metrics such as number of total counts or features detected. However, Seurat is limited as it does not integrate other quality control algorithms, such as doublet detection and ambient RNA detection. SCTK-QC aims to create a standardized, robust workflow which contains a wide

variety of QC algorithms. The following sentences have been added to the discussion to emphasize this point:

“While packages such *Seurat* or *Scanpy* offer the ability to calculate some basic QC metrics and remove poor quality cells, they do not have a comprehensive workflow that includes multiple algorithms for detection of doublets or estimation of contamination from ambient RNA. Additionally, no other packages include the ability to produce HTML reports for easier assessment of quality metrics.”

What with export to Seurat or Monocle proof-objects?

Response 1.9. In our updated version of SCTK-QC, we have added an option to import and export results as a *Seurat* object (in addition to an *AnnData* object). As stated on the *Monocle* website, the previous iteration is becoming deprecated, and the current iteration is in beta testing. As soon as Monocle’s next data structure is in the production version of the package, we will incorporate the ability to convert and export the *SingleCellExperiment* object to the *Monocle* object.

Shiny tools are nice, but do require RAM/ CPU/... what are the requirements for your tools scaled to number of cells?

Response 1.10. To answer this question, we have performed a benchmarking of the QC algorithms using the R/Shiny GUI with different size datasets. One tool, *DoubletFinder*, was memory intensive and will likely not be able to be run with the GUI on large datasets. However, the other tools were able to be run within memory and time requirements that are found on many standard laptops. The following text has been added in the “R/Shiny user interface” section:

“Benchmarking of the Shiny GUI was conducted on a Linux machine using example datasets of 34k and 68k PBMCs. Due to one of the doublet detection algorithms, *DoubletFinder*, requiring large amounts of memory, benchmarking was performed with and without executing *DoubletFinder*. When *DoubletFinder* was applied, the 34k and 68k PBMC dataset required 18.5GB and 4200 seconds, and 38.5GB and 6820 seconds, respectively. When *DoubletFinder* was removed from the list of algorithms executed, the 34k and 68k PBMC dataset required 6GB and 2350 seconds, and 7GB and 3960 seconds, respectively. (Supplementary Table 2). Therefore, all of the QC tools with the possible exception of *DoubletFinder* can be applied on larger datasets using the Shiny GUI.”

In the processing tools, Alevin compatibility would be great, since this is used more and more.

Response 1.11. Thank you for the suggestion. We have added the ability to import data from the Alevin pipeline. This is now mentioned in the following text:

“SCTK-QC can automatically import data from a variety of preprocessing tools and file formats. Supported preprocessing tools include CellRanger, BUStools, STARSolo, SEQC, Optimus, Alevin, and dropEST.”

Comparison to other tools: Seurat itself might not be able to import/export, but many tools are foreseen in Seurat-disk.

Response 1.12. Currently, all Seurat objects are converted to *SingleCellExperiment* objects with data stored on-disk which will be sufficient for datasets under 100K cells in most settings. In the future, the new *scImport* package (currently under development) will allow for import of multi-modal single cell data to on-disk objects and will be supported using HDF5 or TileDB wrapped in the DelayedArray framework. This framework will allow users to perform common array operations without loading the object in memory and we will include the ability to convert from a Seurat-disk object to an Experiment object with on-disk storage.

Reviewer #2 (Expertise: scRNAseq analysis):

This manuscript describes the SCTK-QC pipeline, an R package able to run and process outputs from a number of single-cell analysis bioinformatics tools. In addition to data standardisation and visualisation, the package comes with tools to filter and prepare single cell transcriptomic data for downstream analysis.

The major claims of the paper are that this tool brings together diverse packages into a single consistent interface, suitable for users who may not be familiar with the complexities of installation and execution of varied command line bioinformatics tools. As the number and size of single cell transcriptomics studies is expanding rapidly at the moment, I agree with the authors that this is a worthwhile aim and of importance for the research community. Data quality control and filtering is important for any research project, but it is especially critical for single cell data. By making such analysis more accessible and user-friendly, more research projects are likely to adhere to best-practices for their analysis.

I thought that the manuscript text was very well written - clear and accessible for newcomers to the field. I have almost no comments on the text itself, I thought it was excellent.

My main concern is the difference between the SingleCellTK R package and the SCTK-QC pipeline. I understand that the paper is about the -QC pipeline and not the R package, but I think that the manuscript would benefit from clearer disambiguation between the two.

Response 2.1. Thank you very much for the feedback. The main functions used in the SCTK-QC pipeline are housed in the *singleCellTK* R package. We focused on the “pipeline” aspect for the majority of the manuscript as it can be run using the Docker image on the command line and thus can be incorporated into standardized cloud-based preprocessing workflows. We have added the following sentence to the final paragraph of the Introduction section for clarification:

“The SCTK-QC pipeline is an extension of the *singleCellTK* package and designed as a standalone script which can be run on the command line and incorporated into standard preprocessing pipelines.”

I tried to look into the code used for the paper, but the supplied URL (<https://github.com/campbio/Manuscripts/SCTK-QC>) gave a 404 not accessible error. I guess that this repository is currently set to private.

Response 2.2. We apologize for this oversight. The example code, which was used for reproducing Figure 4 and Table 3 of the manuscript, has now been uploaded to the GitHub repository.

I am not a very experienced user of R so consider myself well positioned to try out the author’s claim of user-friendliness of their package. I attempted to install and run the software myself, but found the tool documentation very difficult to navigate. It is quite spread out (GitHub readme, a GitHub repo wiki, the singleCellTK homepage and the Docker Hub readme - all seem to contain different information). It would be good if one was chosen and the others cut down to just point to that canonical link.

Response 2.3. Thank you very much for the suggestion. We have updated, clarified, and consolidated the SCTK-QC documentation, which is now available at https://camplab.net/sctk/v2.4.1/articles/articles/cmd_qc.html. All other documentation and vignettes can be found at <https://camplab.net/sctk/>.

The website navigation for the singleCellTK website (<https://www.sctk.science>) also seems to be broken - some pages do not exist (under Command Line Analysis dropdown), the navigation changes depending on what page you are on (Command Line Analysis > Batch Correction) and a number of nav links only work from the homepage (following from other pages, articles/ keeps getting added to the URL). Some content I could only find by taking specific paths through the website by trial and error, such as the Shiny UI Reference. Some documentation also seemed out of sync between GitHub and the website (eg. the difficult to find https://github.com/comphiomed/singleCellTK/blob/master/exec/SCTK_runQC_Documentation2.md and https://www.sctk.science/articles/run_qc.html).

Response 2.4. We have updated and improved the documentation in several additional ways. First, we have fixed the missing pages and broken links. Second, we have consolidated the R console analysis and the Shiny analysis for specific steps to a single page. For example, the Import, QC, and Filtering steps can be found at the following URLs:

https://camplab.net/sctk/v2.4.1/articles/articles/cmd_qc.html and <https://camplab.net/sctk/v2.4.1/articles/articles/filtering.html>

The documentation itself is generally extensive but again suffers from a lack of clarity between the different subtools. It would be good to have an immediate introduction to the different entry points for an end user at the start of the docs - R session / command line interface / Docker / Shiny. In fact I couldn't find any documentation about running the Shiny UI anywhere except for the mention of the `singleCellTK()` function in the manuscript. It would be great if the Shiny app could also be bundled into a Docker image to make it faster and easier to run.

Response 2.5. We have clarified the documentation for installation of the package (<https://camplab.net/sctk/v2.4.1/articles/articles/installation.html>) and added documentation for launching the Shiny app (https://camplab.net/sctk/v2.4.1/articles/articles/ui_qc.html). As mentioned previously, the SCTK-QC documentation has been consolidated and clarified within a single page (https://camplab.net/sctk/v2.4.1/articles/articles/cmd_qc.html)

I tried following the installation instructions for the `singleCellTK` package in R on a relatively new system and although it has run smoothly so far it is incredibly slow (it's been going for nearly three hours now as I write this). As such I tried to use Docker image instead but the command given (`docker pull campbio/sctk_qc`) fails because it defaults to the `:latest` tag which does not exist. Running the latest tag (`docker pull campbio/sctk_qc:2.0.0`) does work. There is no Dockerfile for the image at Docker Hub or in GitHub so it wasn't entirely clear to me what the image contained. I tried to run the container in the hope that I could test the `singleCellTK` package within it but it failed with an error (`Error in strsplit(opt[["preproc"]], ",") : non-character argument`). As a last stab I tried the example docker command on an empty directory to see how far it would get but I again got a loading error (`Error in match(x, table, nomatch = 0L) : 'match' requires vector arguments`).

Response 2.6. This is an important point for us to clarify in the documentation. The entry point of the `sctk_qc` docker image is to run the SCTK-QC command line pipeline by default. Therefore, the pipeline expects several arguments specifying the input and output directories, behaviour of the pipeline, etc. Running the docker image without these arguments will generate the error code that is shown. To enter the R console, you can run the docker image with the following command: `docker run --rm -it --entrypoint='/usr/bin/R' campbio/sctk_qc:2.4.1`. You might also want to mount the data folder with `-v` command so that the docker image can access the data. These instructions to run the docker image are now also available on https://camplab.net/sctk/v2.4.1/articles/articles/cmd_qc.html, under "Running SCTK-QC with Docker".

Given this plus the lack of the manuscript repo code to follow I have now given up trying to run the tool myself - either the command line tools, the Shiny app or the R functions. I recommend that the authors try to get a few people outside of their normal community to run the tool as I did using only the available documentation, to try to iron out these problems. I suspect that most will be quite trivial to solve. Apologies if I missed something obvious. In conclusion, I think that the manuscript is excellent and the aim and example outputs of the tool are superb. I think that greater clarity is required between the different components of this package, especially if this

paper is about only part of the total singleCellTK suite. If the QC pipeline is to be successful in targeting computationally inexperienced users then the documentation needs quite a bit of work.

Response 2.7. We hope that the improved documentation will allow users to more easily access the tools via the command line, R console, and Shiny app. We have conducted additional tests on Windows, MacOS, and Linux with different users to ensure that the package can be installed on multiple platforms. Lastly, we agree that installation can be cumbersome and error-prone on different systems due to the large number of dependencies. We are exploring various strategies to allow users to directly access the Shiny UI from the web without having to install the package locally. We have set up a test instance on Amazon Web Services (AWS) which can be accessed and tested with the following link:

<http://3.18.225.121:3838/>

We are also testing Google Cloud Platform with ShinyProxy and several other configurations. Once an optimal platform has been identified, we will make the toolkit available via a standard domain name (e.g. camplab.net/sctk/launch). With the deployment of the interactive tool on the web, we hope that this will further increase the accessibility of scRNA-seq QC and analysis for users without a strong programming background.

Reviewers' Comments:

Reviewer #1:

Remarks to the Author:

The authors did a big effort to improve their manuscript, documentation and tool. It is more clear now where they differ from other available tools in the field and the installation went fluently. A small remark, maybe it is better to first let the user install the python dependencies, since restarting R after installation of these is needed.

I hope that the authors keep track of new advancements in the single cell field by updating their package accordingly.

Reviewer #2:

Remarks to the Author:

I'd like to thank the authors for their comprehensive response and overhaul of the tool documentation. The docs are vastly improved and now work in a stable and predictable manner. They're much easier to navigate and I'm happy that they contain all of the information that I need to run the tools. I found the new online Shiny app to be very slow, but the effort is still appreciated and I'm sure that offering an online service like this is a great way to lower the entry barrier for new users.

I have no further comments about the manuscript or tool.

Phil Ewels

Boston University School of Medicine

March 02, 2022

Dear Editors,

Thank you for the opportunity to revise our manuscript (NCOMMS-21-03980B), "*Comprehensive generation, visualization, and reporting of quality control metrics for single-cell RNA sequencing data*". We appreciate the careful review and constructive comments. We believe that the manuscript is essentially improved after making the suggested edits.

Following this letter are the brief summary of the manuscript and the reviewers' comments with our responses. Changes made in manuscript are marked using track changes.

Sincerely,
Joshua D. Campbell, Ph.D.
Boston University School of Medicine
72 East Concord Street, E-604B
Boston, MA 02218

Brief Summary

Quality control (QC) is a crucial step in single-cell RNA-seq data analysis. The SCTK-QC pipeline generates and visualizes a comprehensive set of QC metrics to streamline the process of detecting and removing poor quality cells and other artifacts.

Twitter: @camplab1

Reviewer #1 (Remarks to the Author):

The authors did a big effort to improve their manuscript, documentation and tool. It is more clear now where they differ from other available tools in the field and the installation went fluently. A small remark, maybe it is better to first let the user install the python dependencies, since restarting R after installation of these is needed.

I hope that the authors keep track of new advancements in the single cell field by updating their package accordingly.

Thank you for the remark about the installation of the Python dependencies. We have updated installation documentation to suggest installing Python dependencies before starting or noting that R will need to be restarted if Python dependencies are installed using Reticulate. The major benefit of the SCTK-QC framework is that it allows us to easily integrate new methods in the pipeline. For example, we are in the process of adding tools for downstream analysis including clustering, trajectory inference, cell type deconvolution.

Reviewer #2 (Remarks to the Author):

I'd like to thank the authors for their comprehensive response and overhaul of the tool documentation. The docs are vastly improved and now work in a stable and predictable manner. They're much easier to navigate and I'm happy that they contain all of the information that I need to run the tools. I found the new online Shiny app to be very slow, but the effort is still appreciated and I'm sure that offering an online service like this is a great way to lower the entry barrier for new users.

I have no further comments about the manuscript or tool.

Phil Ewels

Thank you very much for testing the Shiny app. We agree that the AWS instance was suboptimal. We will soon be releasing a version hosted by the Boston University Shared Computing Cluster which is more stable and substantially faster.